# Experimental Investigation and Mechanism Analysis of Direct Aqueous Mineral Carbonation Using Steel Slag

**Fuxia Zhu [1], Longpeng Cui [1], Yanfang Liu [1], Liang Zou [1], Jili Hou [1], Chenghao Li [1], Ge Wu [1], Run Xu [1], Bo Jiang [2,*] and Zhiqiang Wang [1,*]**

[1] Department of Coal and Syngas Conversion, Sinopec Research Institute of Petroleum Processing Co., Ltd., Beijing 100083, China; zhufuxia.ripp@sinopec.com (F.Z.); cuilongpeng.ripp@sinopec.com (L.C.)
[2] Department of Environmental Science and Engineering, University of Science and Technology Beijing, Beijing 100083, China
* Correspondence: jiangbo_seee@ustb.edu.cn (B.J.); wangzhiqiang.ripp@sinopec.com (Z.W.); Tel.: +86-010-82368211 (B.J.); +86-010-62332867 (Z.W.)

**Abstract:** The carbonation of industrial calcium-rich byproducts such as steel slag demonstrates significant potential for $CO_2$ sequestration. This technique aids in reducing carbon emissions while also promoting waste recycling. Despite its advantages, gaps remain in the understanding of how steel slag characteristics and operational parameters influence the carbonation process, as well as the underlying mechanism of direct aqueous carbonation. We evaluated the carbonation performance of three types of steel slag at temperatures below 100 °C. The slag with the highest $CO_2$ sequestration capacity was chosen for a systematic evaluation of the effects of operating conditions on carbonation efficiency. Thermodynamic analysis indicated that the reactivity of CaO and $Ca(OH)_2$ with $CO_2$ exceeded that of $CaO \cdot SiO_2$ and $2CaO \cdot SiO_2$. Under conditions of 85 °C, a particle size less than 75 μm, an initial $CO_2$ pressure of 0.5 MPa, a liquid-to-solid ratio of 5 mL/g, and a stirring speed of 200 rpm, the steel slag achieved a sequestration capacity ($K$) of 283.5 g($CO_2$)/kg and a carbonation efficiency ($\zeta_{Ca}$) of 51.61%. Characterization of the slag before and after carbonation using X-ray diffraction, SEM-EDS, thermogravimetric analysis, and Fourier transform infrared spectrometry confirmed the formation of new carbonates. Mechanistic analysis revealed that the rate-limiting step initially involved the mass transfer of $CO_2$, transitioning to $Ca^{2+}$ mass transfer as time progressed. Our research provides a viable technique for $CO_2$ capture and a beneficial approach for reutilizing waste steel slag. Furthermore, solid residues after capturing $CO_2$ have the potential for conversion into carbon-negative building materials, offering a sustainable strategy for steel companies and other enterprises with high carbon emissions.

**Keywords:** steel slag; direct aqueous carbonation; $CO_2$ sequestration; parameter optimization; mechanism analysis

## 1. Introduction

Since the beginning of the Industrial Revolution, human production activities have predominantly relied on fossil fuels for energy provision. The combustion of these fossil fuels results in the significant generation of carbon dioxide ($CO_2$), which plays a notable role in amplifying the greenhouse effect [1,2]. This acceleration of the greenhouse effect has garnered substantial attention from governments worldwide due to its adverse effects on the ecological environment, making the reduction of $CO_2$ emissions a pressing global concern. According to statistics, China is responsible for as much as 50% of global carbon emissions, solidifying its position as the top emitter in the world [3]. As the largest developing nation globally, China's rapid industrialization has led to an increasing consumption of fossil fuels. Because clean energy sources such as solar, wind, and hydrogen are not yet feasible alternatives, fossil fuels continue to predominate [4]. Given this critical situation, China's implementation of a "dual carbon" strategy holds significant importance, and there

is an urgent need for the development and research of technologies aimed at reducing $CO_2$ emissions.

The steel industry is a sector characterized by high energy consumption, accounting for approximately 15% of China's total carbon emissions and standing as one of the primary sources of $CO_2$ [5]. During the steel-making process, solid waste, referred to as steel slag, is produced at a rate of 10–15% of the steel output [6]. As of 2021, the global crude steel production reached 1.951 billion tons, with China's share constituting more than 50% of the global total since 2017 and showing a continuous upward trajectory [7,8]. Despite the substantial volume of steel slag generated in China, its effective utilization rate remains below 30%, which is significantly lower than that of other developed nations. This excessive accumulation of steel slag not only leads to resource waste but also results in land occupation and environmental degradation [9]. Research has demonstrated that steel slag possesses promising capabilities for the capture and storage of $CO_2$. Through the mineral carbonation of steel slag, it is possible to enhance the material's properties by reducing its content of free calcium oxide (f-CaO) and free magnesium oxide (f-MgO) while simultaneously reducing $CO_2$ emissions to mitigate the greenhouse effect [10–12]. Furthermore, $CO_2$-captured steel slag can be transformed into carbon-negative building materials, thus promoting sustainable development within the steel industry and other high-carbon-emitting sectors.

Mineral carbonation of steel slag for $CO_2$ sequestration is considered one of the most promising green technologies for achieving the strategic objectives of industrial "dual carbon". Mineral carbonation processes can be categorized into indirect and direct carbonation. Indirect carbonation entails the extraction of Ca/Mg from steel slag using a leaching agent, followed by solid-liquid separation to obtain a solution rich in calcium and magnesium ions, which subsequently undergo carbonation reactions with $CO_2$ to produce $CaCO_3$ and $MgCO_3$ [13]. Although this method can yield high-purity carbonate products, it is plagued by extended leaching reaction times, complex process flows, and substantial energy consumption for leaching agent regeneration and recycling, all of which hinder its large-scale industrial application.

Direct carbonation can be further classified into dry and methods. In the dry method, a gas-solid phase reaction occurs wherein $CO_2$ gas diffuses into the steel slag and reacts with its active components. However, the dense structure of steel slag impedes $CO_2$ diffusion, resulting in sluggish carbonation rates even under elevated temperatures and initial $CO_2$ pressure conditions. These low conversion rates render it unsuitable for industrial-scale applications [14,15]. Santos et al. [16] investigated the dry method carbonation of steel slag and found that under conditions of a reaction temperature of 500 °C, $CO_2$ concentration of 75%, initial $CO_2$ pressure of 3 bar, and reaction time of 50 min, the sequestration capacity and carbonation efficiency were 83.8 g($CO_2$)/kg and 29%, respectively. Ghouleh et al. [17] determined that reaction time and temperature are the primary factors influencing the carbonation performance of steel slag. At 650 °C and an initial $CO_2$ pressure of 20 bar, they achieved a maximum carbonization conversion of 26%, equivalent to a capacity of 120 g($CO_2$)/kg.

In the wet method, a gas-liquid–solid three-phase system is present; $CO_2$ dissolves in water to form carbonic acid, and steel slag gradually dissolves in a weakly acidic solution, subsequently precipitating as carbonates upon reacting with bicarbonate ions [18]. This method demonstrates favorable kinetic properties under lower temperatures and initial $CO_2$ pressures, necessitating reduced energy input and thereby augmenting its economic feasibility. The majority of the studies involving steel slag for $CO_2$ mineralization primarily focus on the wet method of direct carbonation. Ibrahim et al. [19] employed response surface methodology to investigate the effects of initial $CO_2$ pressure on the aqueous carbonation reactions of steel slag. The results indicate that within a specific range, decreasing the solid-to-liquid ratio and increasing the pressure can enhance $CO_2$ sequestration. Under optimal reaction conditions, the $CO_2$ sequestration capacity ($K$) reached 283 g($CO_2$)/kg, accompanied by a carbonation efficiency ($\zeta_{Ca}$) of 67%. Chang et al. [20] utilized a rotating

packed bed to examine the aqueous carbonation of steel slag, revealing that the most influential operational parameter affecting carbonation kinetics was the reaction temperature. Under optimal conditions of 65 °C, 750 rpm, and t = 30 min, a $CO_2$ sequestration capacity of 404.8 g($CO_2$)/kg and a carbonation efficiency of up to 93.5% were achieved. Furthermore, He et al. [21] employed machine learning to model and predict the $CO_2$ sequestration process using steel slag slurry and to investigate the effect of process parameters and slag composition on $CO_2$ sequestration. However, current research primarily centers on the effects of particle size, temperature, reaction time, $CO_2$ concentration, and pressure on carbonation performance, while the underlying mechanism of carbonation remains incompletely understood and necessitates further investigation.

Building upon the aforementioned analysis, this study initiates an exploration into the carbonation performance of various types of steel slag at low temperatures (<100 °C). Through the characterization of samples before and after carbonation, the reaction mechanism of steel slag carbonation is elucidated, and, from a thermodynamic standpoint, the reactivity of diverse calcium-based components in steel slag toward $CO_2$ is analyzed. Subsequently, the study explores the effects of particle size (*D*), temperature (*T*), initial $CO_2$ pressure (*p*), liquid-to-solid ratio (L/S), and rotational speed (*r*) on the carbonation performance of steel slags. A comprehensive investigation into the mechanism of $CO_2$ capture within a reaction system employing steel slag is also undertaken. This study optimized the optimal process parameters and clarified the reaction mechanism for the mineral carbonation of steel slag for $CO_2$ sequestration, which lays the foundation for the industrial scale up of this technology. In addition, this research provides a valuable contribution to the reduction of carbon emissions and the enhancement of the resource recycling rate of steel slag, thereby facilitating the development of a circular economy.

## 2. Materials and Methods

### 2.1. Materials

The steel slag samples utilized in this investigation were sourced from three steel mills situated in Shandong Province, China, and were designated SS-1, SS-2, and SS-3. These samples underwent an initial drying process at 105 °C until a constant weight was achieved, followed by sieving to obtain samples of varying particle sizes: >180 μm, 180~150 μm, 150~120 μm, 120~75 μm, and <75 μm. Table 1 presents the chemical composition of the steel slag samples. The pure $CO_2$ employed for the experiments was procured from Beijing Huanyu Jinghui Gas Technology Co., Ltd. (Beijing, China), with a volume fraction of 99.9%.

**Table 1.** Chemical composition of the different steel slag samples.

| Sample | wt (%) | | | | | | | | | |
|--------|--------|--------|--------|--------|--------|--------|--------|--------|--------|--------|
| | $SiO_2$ | $Al_2O_3$ | $Fe_2O_3$ | CaO | MgO | $TiO_2$ | $Na_2O$ | $K_2O$ | $P_2O_5$ | $SO_3$ |
| SS-1 | 22.21 ± 0.89 | 1.35 ± 0.04 | 0.39 ± 0.01 | 64.73 ± 2.14 | 6.25 ± 0.18 | 1.07 ± 0.04 | 0.02 ± 0.00 | 0.00 ± 0.00 | 0.00 ± 0.00 | 0.23 ± 0.01 |
| SS-2 | 9.47 ± 0.21 | 1.98 ± 0.07 | 24.44 ± 0.76 | 51.79 ± 2.31 | 5.95 ± 0.25 | 0.71 ± 0.02 | 0.08 ± 0.00 | 0.05 ± 0.00 | 1.45 ± 0.03 | 0.41 ± 0.01 |
| SS-3 | 16.80 ± 0.45 | 4.42 ± 0.10 | 24.90 ± 0.63 | 39.10 ± 1.78 | 4.38 ± 0.32 | 1.00 ± 0.05 | 0.34 ± 001 | 0.40 ± 0.01 | 1.47 ± 0.05 | 1.32 ± 0.03 |

### 2.2. Experimental Section

Figure 1 illustrates a schematic diagram of the experimental setup. In each carbonation experiment, a specific quantity of deionized water and sample were introduced into a high-pressure autoclave reactor. The reactor was sealed and brought to the target temperature. Next, $CO_2$ was introduced into the reactor until the predetermined pressure was achieved, and mechanical stirring commenced with precise timing. Following a reaction period of 2 h, the heating process was terminated, and the reactor was rapidly cooled to room temperature. The suspension in the reactor underwent filtration using 0.7 μm filter paper. The resulting solid fraction was then dried at 105 °C for 24 h. The carbonation performance of the steel slag was quantified through thermogravimetric analysis (TG) conducted on the dried solids. The reaction mechanism was revealed through a series of characterizations

of the steel slag before and after carbonation. The experimental program of this study is presented in Table 2.

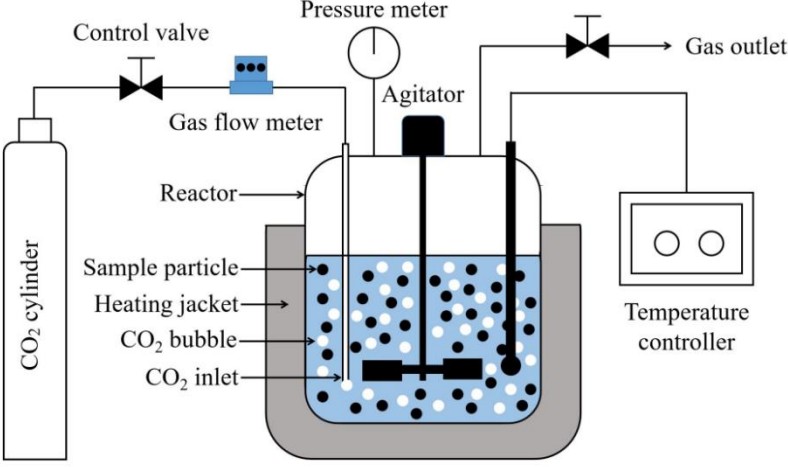

**Figure 1.** Schematic of the experimental apparatus.

**Table 2.** Experimental program of this study.

| Variables | Feedstock | $D$ (µm) | $T$ (°C) | $p$ (MPa) | L/S (mL/g) | $r$ (rpm) |
|---|---|---|---|---|---|---|
| Steel slags | SS-1, SS-2, SS-3 | 120~75 | 65 | 2 | 15 | 200 |
| $D$ | SS-2 | >180, 180~150, 150~120, 120~75, <75 | 65 | 2 | 15 | 200 |
| $T$ | SS-2 | <200 | 25, 45, 65, 85, 105 | 2 | 15 | 200 |
| $p$ | SS-2 | <200 | 105 | 0.1, 0.5, 1.0,1.5, 2 | 15 | 200 |
| L/S | SS-2 | <200 | 105 | 0.5 | 1, 5, 10, 15, 20 | 200 |
| $r$ | SS-2 | <200 | 105 | 0.5 | 15 | 200, 400, 600, 800 |

### 2.3. Calculation of Carbonation Performance

TG analysis is performed using a thermogravimetric analyzer to quantify the weight loss experienced by steel slag at elevated temperatures, allowing for the precise determination of the quantity of $CO_2$ sequestered by the steel slag. The method entails placing a defined mass of the carbonated steel slag product into the thermogravimetric analyzer and subjecting the sample to heating under a temperature range spanning from 50 °C to 950 °C, at a rate of 10 °C per minute, while maintaining an $N_2$ atmosphere. The temperature was maintained at both 105 °C and 550 °C for 10 min each, followed by a 5-min dwell at 950 °C. Weight losses observed in the temperature ranges of 50–105 °C, 105–550 °C, and 550–950 °C were associated with water evaporation, the decomposition of $Ca(OH)_2$ and $MgCO_3$, and the decomposition of $CaCO_3$, respectively. The quantity of $CO_2$ was calculated using the dry weight of the carbonated sample and its corresponding weight loss was observed within the temperature range of 550 °C to 950 °C, following Equation (1):

$$\omega_{CO_2}[\text{wt\%}] = \frac{\Delta m_{550-950°C}[g]}{m_{105°C}[g]} \times 100\% \tag{1}$$

$K$ and $\zeta_{Ca}$ are employed to assess the degree of the carbonation reaction, computed in accordance with Equations (2) and (3):

$$K[\text{g}(\text{CO}_2)/\text{kg}] = \frac{\omega_{\text{CO}_2}[\text{wt\%}]}{1 - \omega_{\text{CO}_2}[\text{wt\%}]} \times 1000 \tag{2}$$

$$\zeta_{\text{Ca}}[\%] = \frac{\frac{\omega_{\text{CO}_2}[\text{wt\%}]}{1 - \omega_{\text{CO}_2}[\text{wt\%}]} \times \frac{M_{\text{Ca}}[\text{g/mol}]}{M_{\text{CO}_2}[\text{g/mol}]]}}{Ca_{\text{total}}[\text{wt\%}]} \tag{3}$$

where $M_{Ca}$ and $M_{\text{CO}_2}$ represent the molar masses of Ca and $CO_2$, respectively, while $Ca_{total}$ denotes the Ca content in the fresh steel slag samples.

### 2.4. Characterization of the Samples

The samples underwent analysis of their chemical compositions via X-ray fluorescence spectroscopy (XRF, EA1400, Hitachi, Tokyo, Japan). The crystal phases of the samples, both before and after carbonation, were investigated through X-ray diffraction (XRD, PANalytical Empyrean, PANalytical, Malvern, UK), using Cu–Ka radiation operating at 40 kV and 40 mA. The XRD analysis covered a 2θ scanning angle range from 10° to 70°, utilizing a step size of 0.01°. To examine the microstructure of the samples, evaluate the surface elemental distribution, and identify the formation of carbonates, scanning electron microscopy coupled with energy-dispersive spectroscopy (SEM-EDS, Quanta-200, FEI, Hillsboro, OR, USA) was utilized. The testing conditions were 5 kV voltage and 45 μA/cm$^2$ current density. The changes of the principal chemical bonds present in the samples both before and after the carbonation reaction were analyzed using Fourier transform infrared spectroscopy (FTIR, Tensor27, Bruker, Berlin, Germany) over a wavelength range of 4000–400 cm$^{-1}$ at a resolution of 1 cm$^{-1}$. The changes in the specific surface area and pore size distribution of the steel slag samples, both before and after the carbonation, were analyzed via Brunauer Emmett Teller (BET, TRISTAR II 3020, Micromeritics, Norcross, GA, USA) analysis. The sample was pretreated at 300 °C for 4 h, and then was examined at −196 °C. Finally, the particle size distribution of the samples was determined using a laser particle size analyzer (Mastersizer3000, Malvern Panalytical, Malvern, UK).

## 3. Results and Discussion

### 3.1. Carbonation Performance of Different Steel Slags

The sequestration capacities ($K$) for steel slag samples SS-1, SS-2, and SS-3 under identical experimental conditions were determined to be 106.8 ± 3.2, 191.9 ± 3.9, and 136.9 ± 2.6 g(CO$_2$)/kg, respectively. The ability of industrial solid waste to sequester $CO_2$ is not solely contingent on the content of alkaline components but is also influenced by its phase composition [22]. Steel slag primarily comprises four calcium-based active components: $CaO$, $Ca(OH)_2$, $CaO \cdot SiO_2$, and $2CaO \cdot SiO_2$. The reaction equations involving these four components in the carbonation process, along with their corresponding Gibbs free energy changes ($\Delta_r G_m^\theta$, kJ/mol), are presented in Table 3 [23–25]. At a temperature of 65 °C, the Gibbs free energy changes for the reactions between these calcium-based active components and $CO_2$ are as follows: −124.22, −69.19, −38.67, and −57.17 kJ/mol. This indicates that all reactions proceed spontaneously, with the reactivity ranking as follows: $CaO > Ca(OH)_2 > 2CaO \cdot SiO_2 > CaO \cdot SiO_2$.

Figure 2 illustrates the XRD spectra of the three types of steel slag. It is evident that the phase compositions of SS-2 and SS-3 substantially differ from that of SS-1. In SS-2 and SS-3, the primary calcium-based active component is $Ca(OH)_2$, while SS-1 also includes $CaO \cdot SiO_2$ and $2CaO \cdot SiO_2$. Thermodynamic analysis reveals that $Ca(OH)_2$ exhibits a higher reactivity with $CO_2$ as compared to $CaO \cdot SiO_2$ or $2CaO \cdot SiO_2$, which elucidates the superior carbonation performances of SS-2 and SS-3 relative to SS-1. Additionally, as indicated in Table 1, SS-2 boasts higher contents of CaO and MgO at 51.76% and 5.95%, respectively, in contrast to 39.10% and 4.38% in SS-3. Consequently, the carbonation performance of SS-2

surpasses that of SS-3. Subsequent investigations will focus on SS-2 to explore the influence of particle size (*D*), temperature (*T*), pressure (*p*), liquid-to-solid ratio (L/S), and rotational speed (*r*) on the carbonation performance of steel slag.

**Table 3.** Potential carbon sequestration reactions of the main calcium-based active components in steel slag and their Gibbs free energies (atmospheric pressure).

| Phase | Reaction Equation | $\Delta_r G_m^{\theta}$ (kJ/mol) |
|---|---|---|
| CaO | $CaO + CO_2 \rightarrow CaCO_3$ | $-178.32 + 0.16(T + 273.15)$ |
| $Ca(OH)_2$ | $Ca(OH)_2 + CO_2 \rightarrow CaCO_3 + H_2O$ | $-113.15 + 0.13(T + 273.15)$ |
| $CaO \cdot SiO_2$ | $CaO \cdot SiO_2 + H_2O + CO_2 \rightarrow CaCO_3 + SiO_2 \cdot H_2O$ | $-92.77 + 0.16(T + 273.15)$ |
| $2CaO \cdot SiO_2$ | $1/2(2CaO \cdot SiO_2) + 1/2H_2O + CO_2 \rightarrow CaCO_3 + 1/2SiO_2 \cdot H_2O$ | $-111.27 + 0.16(T + 273.15)$ |

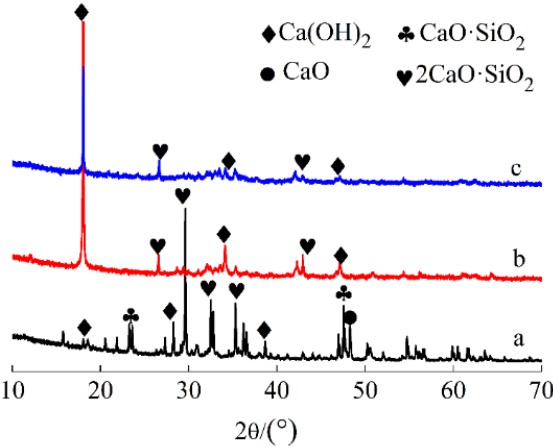

**Figure 2.** XRD spectra of different steel slag samples, (a) SS-1, (b) SS-2, and (c) SS-3.

### 3.2. Characterization of the Samples

### 3.2.1. TG Analysis

Figure 3 illustrates the weight loss curves for the SS-2 steel slag both before and after the carbonation reaction. The unprocessed sample exhibits a noticeable weight loss within the temperature range of 105–550 °C, which corresponds to the decomposition of $Ca(OH)_2$ and indicates the presence of $Ca(OH)_2$ in the original sample. After undergoing the carbonation reaction, the weight loss in the 105–550 °C range notably diminishes, whereas the weight loss between 550–850 °C experiences a significant increase, reaching 21.92%. This observation implies that following the carbonation reaction, $Ca(OH)_2$ reacts with $CO_2$ to yield $CaCO_3$.

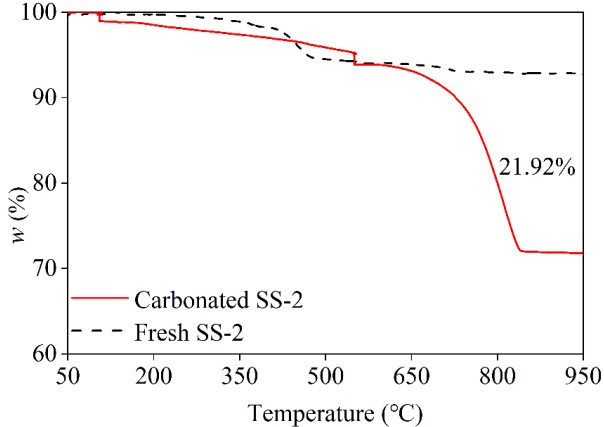

**Figure 3.** Thermal gravimetrical curves for fresh and carbonated SS-2 samples.

### 3.2.2. XRD Analysis

Figure 4 presents the XRD spectra of the SS-2 steel slag both before and after the carbonation reaction. The initial SS-2 steel slag primarily contains $Ca(OH)_2$ and $2CaO \cdot SiO_2$. However, following carbonation, the peak corresponding to $Ca(OH)_2$ vanishes, the peak for $2CaO \cdot SiO_2$ weakens, and the $CaCO_3$ peak intensifies. These findings affirm that $Ca(OH)_2$ in the initial SS-2 steel slag undergoes complete conversion into $CaCO_3$ through carbonation, a conclusion supported by the results of the TG analysis of the carbonated slag. Additionally, the partial carbonation of $2CaO \cdot SiO_2$ into $CaCO_3$ suggests that $Ca(OH)_2$ displays a greater reactivity with $CO_2$ compared to $2CaO \cdot SiO_2$, which aligns with the thermodynamic analysis (Table 3). Although SS-2 has a high Fe content (Table 1), $FeCO_3$ was not observed in the XRD analysis. Carbonates of other minerals like Al, Na, and K were also absent. Moreover, the XRD results also show that the carbonation process yielded negligible quantities of $MgCO_3$. These results indicate that reaction condition was unfavorable for the formation of carbonates by these elements [23].

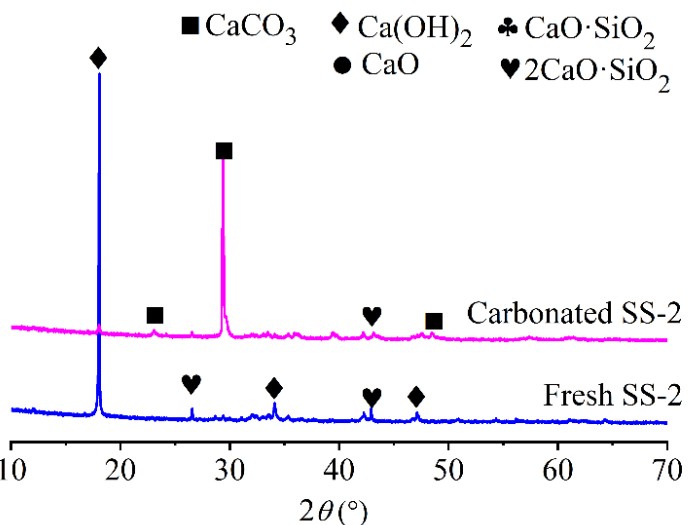

**Figure 4.** XRD patterns of fresh and carbonated SS-2 samples.

### 3.2.3. SEM-EDS Analysis

Figure 5a,b provide an illustration of the crystal morphology of the SS-2 steel slag both before and after the carbonation reaction. The morphology of the initial SS-2 steel slag is characterized by particle aggregation with unevenly distributed pores. Notable morphological transformations occur subsequent to carbonation, including the emergence of numerous cubic particle crystals that aggregate. Complementing after the carbonation energy-dispersive spectroscopy (EDS) analysis (Figure 5c), the composition is unequivocally identified as $CaCO_3$, a finding that corroborates the results of the XRD phase analysis. However, it is important to note that the surface development of $CaCO_3$ appears uneven following the carbonation reaction. This nonuniformity may impede the penetration of unreacted slag into the liquid phase, thereby increasing the difficulty of further reactions between the unreacted Ca-based active phases within the slag and $CO_2$. Consequently, this could limit the slag's $CO_2$ sequestration capacity [10]. Furthermore, freshly carbonated slag comprises particles of varying sizes, exhibiting a surface rich in fine-pore structures. In contrast, carbonated samples create compact aggregates with relatively consistent particle sizes.

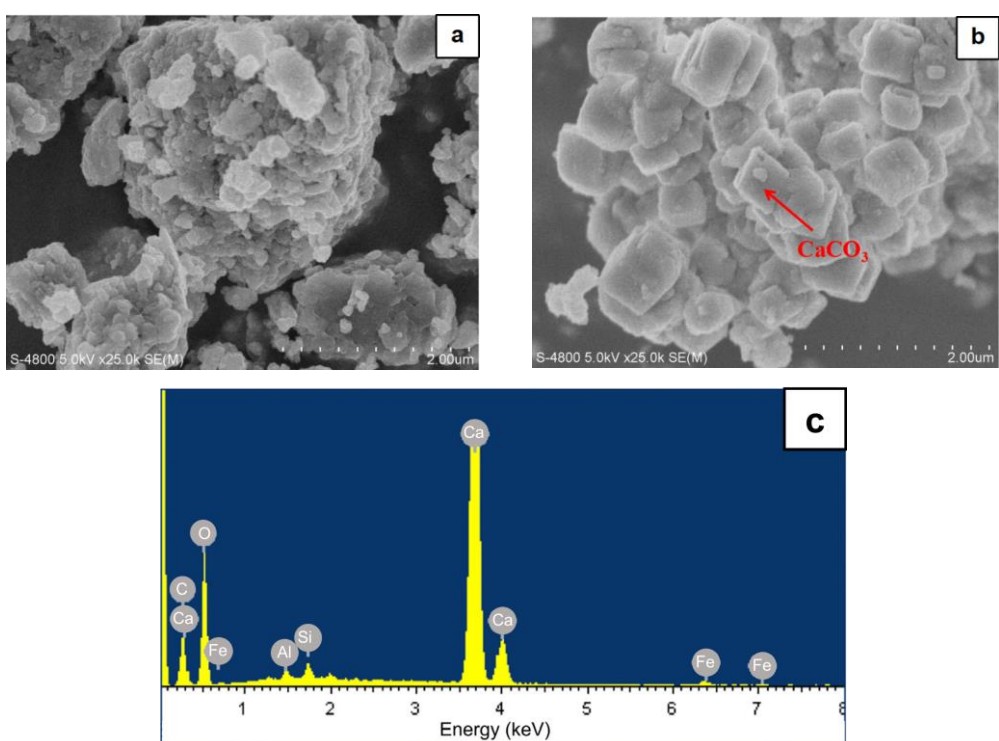

**Figure 5.** SEM-EDS analysis of (**a**) fresh and (**b**,**c**) carbonated steel slags.

### 3.2.4. FTIR Spectra

Figure 6 presents the infrared spectra of the SS-2 steel slag both before and after the carbonation reaction. The spectrum of the initial SS-2 steel slag not only reveals a broad absorption peak at 3422 cm$^{-1}$ but also displays a narrower peak at 3644 cm$^{-1}$. This narrower peak at 3644 cm$^{-1}$ corresponds to the H–O stretching vibrations in Ca(OH)$_2$ [26], thus confirming the presence of Ca(OH)$_2$. This observation aligns with the previous XRD phase analysis and the TG results. Subsequent to the carbonation reaction of the SS-2 steel slag, prominent absorption peaks emerge in the resulting material at 715 cm$^{-1}$, 873 cm$^{-1}$, and 1450 cm$^{-1}$. Specifically, the peak at 715 cm$^{-1}$ corresponds to the in-plane bending vibration of the O–CO- group in CaCO$_3$, the peak at 873 cm$^{-1}$ represents the out-of-plane bending vibration of the O–CO- group, and the peak at 1450 cm$^{-1}$ is associated with the C–O antisymmetric stretching vibrations in CaCO$_3$ [27]. These distinctive features provide compelling evidence that the primary product of the carbonation reaction is indeed CaCO$_3$. In addition, the peak at 873 cm$^{-1}$ may also be attributed to Ca–O [28].

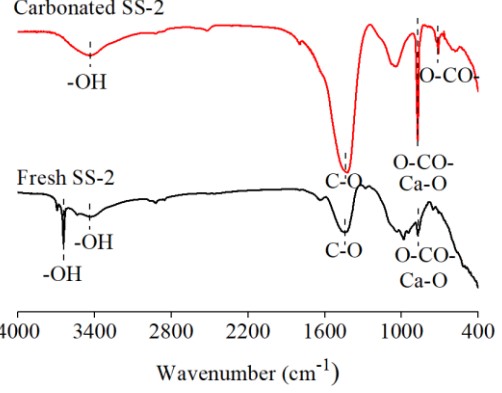

**Figure 6.** FTIR spectra of fresh and carbonated SS-2 samples.

*3.3. Effect of Operational Parameters on the Carbonation of Steel Slag*

3.3.1. Influence of Particle Size

Figure 7 depicts the effect of particle size on direct aqueous carbonation under $T = 65\,°C$, $p = 2$ MPa, L/S = 15 mL/g, and $r = 200$ rpm conditions. It is evident that as the particle size decreases below 180 μm, both the sequestration rate ($K$) and the carbonation rate $\zeta_{Ca}$ exhibit a notable increase. Specifically, $K$ rises from 50.8 g($CO_2$)/kg to 219.5 g($CO_2$)/kg, and $\zeta_{Ca}$ increases from 10.20% to 40.27%. This observed increase can be attributed to the growth in the specific surface area of the steel slag as the particle size diminishes (as detailed in Table 4). Simultaneously, the milled steel slag undergoes a reduction in lattice energy, leading to the generation of lattice dislocations, defects, and recrystallization at sites where lattice energy is lost. These alterations facilitate an expanded contact area between the slag minerals and $CO_2$, thereby enhancing the interaction forces between the minerals and $CO_2$ and accelerating the carbonation reaction [29]. Consequently, the optimal particle size for the direct aqueous carbonation of steel slag is less than 75 μm.

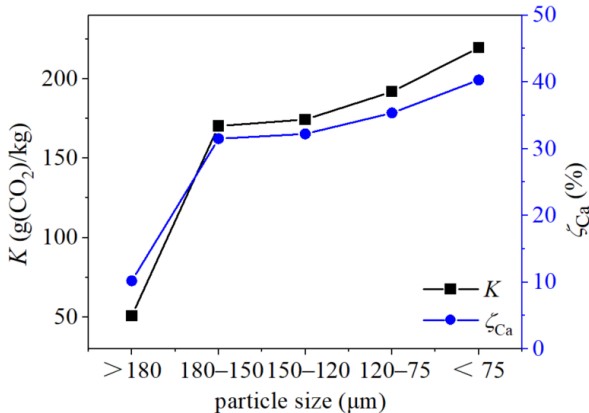

**Figure 7.** Influence of particle size on the carbonation performance of steel slag.

**Table 4.** BET analysis of SS-2 particles of different sizes.

| Sample Parameter | Unit | >180 | 180~150 | 150~120 | 120~75 | <75 |
|---|---|---|---|---|---|---|
| BET surface area | $m^2/g$ | 2.005 ± 0.081 | 10.227 ± 0.113 | 10.966 ± 0.152 | 13.909 ± 0.094 | 14.162 ± 0.172 |
| Total pore volume | $cm^3/g$ | 0.005 ± 0.000 | 0.022 ± 0.001 | 0.024 ± 0.001 | 0.029 ± 0.001 | 0.038 ± 0.001 |

3.3.2. Influence of Reaction Temperature

Figure 8 illustrates the effect of the reaction temperature on the direct aqueous carbonation of steel slag, considering $D < 200$ μm, $p = 2$ MPa, L/S = 15 mL/g, and $r = 200$ rpm. This observation indicates that the carbonation performance of steel slag experiences a decline as the temperature varies from 25 °C to 45 °C. However, a significant enhancement in carbonation performance was noted as the temperature rises from 45 °C to 85 °C, with the sequestration capacity ($K$) increasing from 157.9 g($CO_2$)/kg to 260.7 g($CO_2$)/kg and the carbonation rate $\zeta_{Ca}$ increasing from 29.29% to 47.62%. Beyond this temperature range, only marginal increases in both $K$ and $\zeta_{Ca}$ were noted.

This finding contrasts with our prior research on the effect of temperature on the carbonation efficiency of carbide slag [22]. The disparity may be ascribed to the distinct phase compositions of carbide slag and steel slag. Carbide slag primarily comprises $Ca(OH)_2$, which readily dissociates into $Ca^{2+}$ ions at lower temperatures for reaction with $CO_2$. Below 65 °C, the leaching of $Ca^{2+}$ ions constitutes the rate-limiting step in the reaction. With a further temperature increase, the solubility of $CO_2$ rapidly diminishes, hindering the formation of $CO_3^{2-}$ ions and, subsequently, the precipitation of $CaCO_3$, making $CO_2$ dissolution the new rate-limiting step [19].

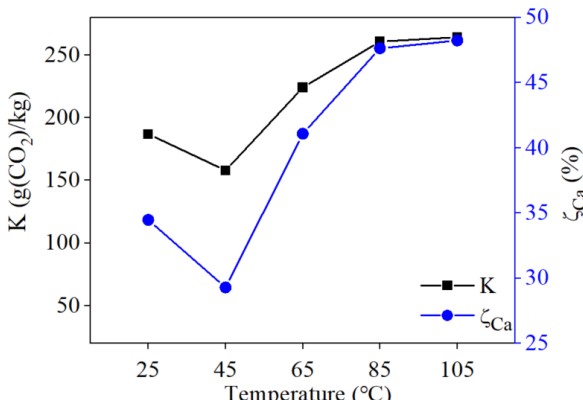

**Figure 8.** Influence of the reaction temperature on the carbonation performance of steel slag.

In contrast, steel slag possesses a more complex phase composition, including both $Ca(OH)_2$ and $2CaO \cdot SiO_2$. The $Ca^{2+}$ ions in $2CaO \cdot SiO_2$ are more challenging to dissociate, necessitating higher temperatures for effective leaching. Therefore, below 45 °C, $CO_2$ dissolution serves as the rate-limiting step. As the temperature increases, $Ca^{2+}$ leaching becomes the rate-limiting step for carbonation. The leaching rate of $Ca^{2+}$ accelerates, and the amount leached increases within a certain time frame, thus promoting $CaCO_3$ formation [30]. Consequently, the optimal temperature for the direct aqueous carbonation of steel slag was found to be 85 °C.

### 3.3.3. Influence of Initial $CO_2$ Pressure

Figure 9 illustrates the effect of the initial $CO_2$ pressure on the direct aqueous carbonation of steel slag under $D < 200$ μm, $T = 105$ °C, L/S = 15 mL/g, and $r = 200$ rpm conditions. As observed, when the pressure is below 0.5 MPa, both $K$ and $\zeta_{Ca}$ increase significantly with the rising pressure. The $K$ value escalates from 166.9 g($CO_2$)/kg to 214.8 g($CO_2$)/kg, while the carbonation rate $\zeta_{Ca}$ improves from 30.88% to 39.42%. However, beyond the threshold of 0.5 MPa, a plateau is reached in both $K$ and $\zeta_{Ca}$ upon increasing the pressure to 2.0 MPa. Consequently, the optimal initial $CO_2$ pressure for the direct aqueous carbonation of steel slag was determined to be 0.5 MPa.

$$[CO_2] = k_{co_2} \times P(CO_2) \tag{4}$$

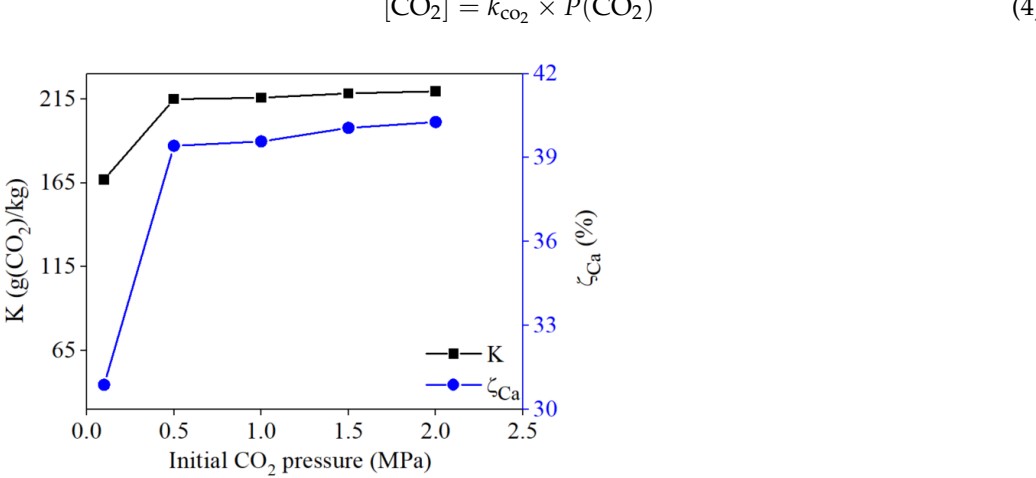

**Figure 9.** Effect of the initial $CO_2$ pressure on the carbonation performance of steel slag.

The influence of the initial $CO_2$ pressure on the $CO_2$ sequestration capacity of steel slag can be explained with Henry's Law (Equation (4)). [$CO_2$] represents the concentration of $CO_2$ in water, while $k_{CO_2}$ is Henry's constant and $P(CO_2)$ is the partial pressure of $CO_2$ gas. At a constant temperature, the solubility of $CO_2$ molecules increases proportionally

with pressure, indicating that a higher initial $CO_2$ pressure leads to the increased dissolution of $CO_2$ in the slurry. This enhanced dissolution facilitates the formation of $CaCO_3$ precipitates [14,31].

For steel slag, when $P(CO_2) < 0.5$ MPa, the dissolution of $CO_2$ serves as the rate-limiting step, and both the sequestration capacity ($K$) and the carbonation rate $\zeta_{Ca}$ primarily depend on the concentration of $CO_3{}^{2-}$ ions. As the initial $CO_2$ pressure increases, the concentration of $CO_2$ and, consequently, the generated $CO_3{}^{2-}$ ions in the slurry rise, resulting in an increase in both $K$ and $\zeta_{Ca}$. When $P(CO_2) > 0.5$ MPa, the leaching of $Ca^{2+}$ ions from the solid matrix becomes the rate-limiting step. At this juncture, the $CO_2$ concentration in the slurry has already reached saturation and no longer influences the carbonation efficiency. Instead, the intrinsic properties of the raw material govern the carbonation performance [32].

### 3.3.4. Influence of the Liquid-to-Solid Ratio

Figure 10 depicts the effect of the liquid-to-solid ratio on the direct aqueous carbonation of steel slag under specific conditions: a constant reaction temperature of 105 °C, a particle size less than 200 μm, an initial $CO_2$ pressure of 0.5 MPa, and a stirring speed of 200 rpm. It is evident that when the liquid-to-solid ratio is less than 5 mL/g, both the sequestration capacity ($K$) and the carbonation rate $\zeta_{Ca}$ increase significantly. Specifically, $K$ escalates from 98.2 g($CO_2$)/kg to 217.6 g($CO_2$)/kg, while the carbonation rate $\zeta_{Ca}$ rises from 10.73% to 39.92%. However, with further increases in L/S, there is a decline in both $K$ and $\zeta_{Ca}$. The optimal liquid-to-solid ratio for the direct aqueous carbonation of steel slag was identified as 5 mL/g.

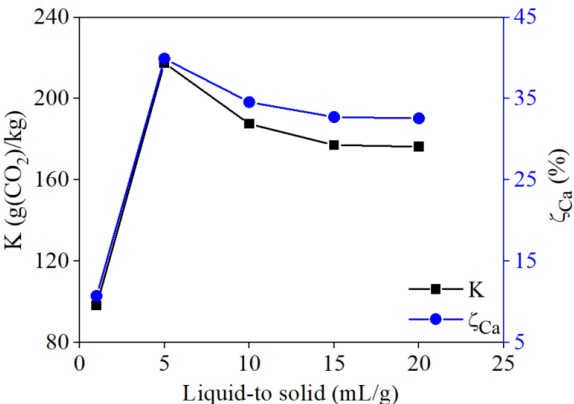

**Figure 10.** Influence of the liquid-to-solid ratio on the carbonation performance of steel slag.

The direct aqueous carbonation process of steel slag is closely linked to the diffusion and dissolution of $CO_2$ in water, as well as the dissolution of alkaline substances in steel slag. Prior to reaching a liquid-to-solid ratio of 5 mL/g, water acts as the medium that facilitates the reaction between the solid matrix and gaseous $CO_2$. However, the leaching of $Ca^{2+}$ ions from the solid matrix is insufficient, limiting the extent of the reaction. As the liquid-to-solid ratio continues to increase, excess water in the reaction system impedes the permeation of $CO_2$ due to capillary forces. This hindrance inhibits the contact between $CO_2$ and active reaction sites, resulting in incomplete carbonation. Consequently, the carbonation efficiency of steel slag decreases [33].

### 3.3.5. Influence of Rotational Speed

Figure 11a displays the influence of rotational speed on the direct aqueous carbonation of steel slag under $T = 105$ °C, $D < 200$ μm, $p = 0.5$ MPa, and L/S = 15 mL/g conditions. As the rotational speed increases from 200 rpm to 800 rpm, both $K$ and $\zeta_{Ca}$ experience a minor increase, indicating that rotational speed has a marginal effect on the carbonation performance of steel slag. However, as shown in Figure 11b, elevating the rotational speed

enhances the reaction rate of the direct aqueous carbonation of steel slag. This enhancement may be attributed to an increased contact area or a thinner diffusion layer, facilitating better interaction among the gas, liquid, and solid phases [34]. Additionally, higher stirring rates might induce erosive effects on the particles, improving the reactivity of the solid matrix surface and preventing the newly formed $CaCO_3$ precipitate from coating unreacted sample surfaces [35].

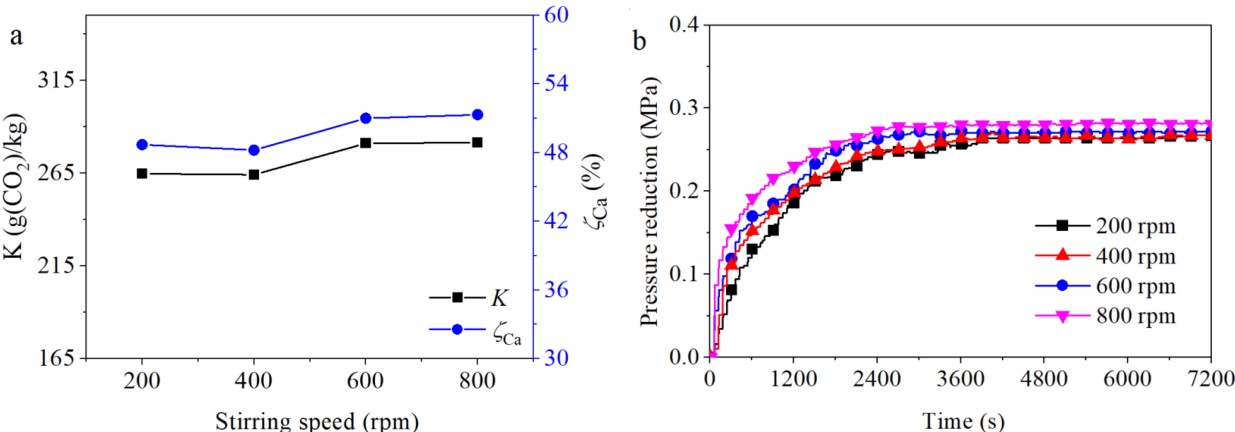

**Figure 11.** (**a**) Effect of the rotational speed on the carbonation performance of steel slag and (**b**) carbonation pressure reduction over time at various stirring speeds.

### 3.4. Mechanism Analysis of Steel Slag for $CO_2$ Sequestration

The chemical reactions involved in the direct aqueous carbonation of steel slag include [36]:

I.    Dissolution of $CO_2$ in the liquid phase to form $H_2CO_3$ ($H_2O + CO_2 \rightarrow H_2CO_3$);

II.    Dissociation of $H_2CO_3$ to form $HCO_3^-$ or $CO_3^{2-}$ ions ($H_2CO_3 \rightarrow HCO_3^- + H^+$; $HCO_3^- \rightarrow CO_3^{2-} + H^+$);

III.    Irreversible hydration of CaO in steel slag ($CaO + H_2O \rightarrow Ca(OH)_2$), dissolution and ionization of $Ca(OH)_2$ ($Ca(OH)_2 \rightarrow Ca^{2+} + 2OH^-$);

IV.    Reaction between $CO_3^{2-}$ and $Ca^{2+}$ ions to form $CaCO_3$ ($Ca^{2+} + CO_3^{2-} \rightarrow CaCO_3$).

The process of the direct aqueous carbonation of steel slag occurs in two stages, as depicted in Figure 12. In the first stage, $CO_2$ molecules diffuse from the gas phase into the gas film and then to the gas-liquid interface, eventually melting into the liquid phase. This diffusion process follows the dual-film theory. The active components in steel slag diffuse from the solid interior to the liquid film at the solid-liquid interface, a process illustrated by Fick's law. Carbonation between $CO_3^{2-}$ and $Ca^{2+}$ ions initiates in the liquid film close to the gas-liquid interface, where unreacted calcium-based active components in steel slag particles gradually shrink [37]. The formed $CaCO_3$ is primarily in the liquid phase, with a minor fraction coating the unreacted steel slag particles. The diffusion rate of $Ca^{2+}$ ions is relatively higher than that of $CO_2$, keeping the concentration of $Ca^{2+}$ ions constant in the liquid phase. Therefore, the rate of the carbonation reaction is governed by the mass transfer of $CO_2$ between the gas and liquid phases.

In the second stage, $CO_2$ molecules diffuse from the liquid film to the liquid film at the solid-liquid interface, and the carbonation reaction between $Ca^{2+}$ and $CO_3^{2-}$ ions also shifts to this film. Simultaneously, unreacted components in steel slag particles continue to shrink. The diffusion rate of $Ca^{2+}$ ions decreases, resulting in a lower concentration of $Ca^{2+}$ ions in the liquid phase. The formed calcium carbonate primarily coats the steel slag particle surfaces, impeding further dissolution of $Ca^{2+}$ ions and reducing the rate of the carbonation reaction. Consequently, the mass transfer of $Ca^{2+}$ ions between the solid and liquid phases controls the reaction rate [38].

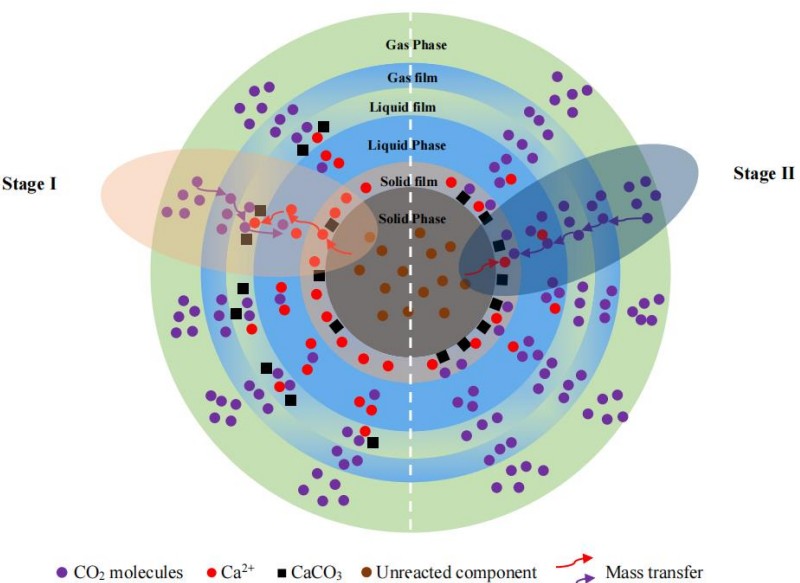

**Figure 12.** Mechanism analysis of the direct aqueous mineral carbonation of steel slag.

## 4. Conclusions

(1) Particle size, temperature, pressure, and liquid-to-solid ratio significantly influence both the sequestration rate $K$ and carbonation rate $\zeta_{Ca}$ of steel slag, while the influence of rotational speed is minor. The optimal carbonation performance of steel slag is observed under the following conditions: $D < 75$ µm, $T = 105\ °C$, $p = 0.5$ MPa, and L/S = 5 mL/g. Under these conditions, the sequestration and carbonation rates reached 283 g($CO_2$)/kg and 51.61%, respectively.

(2) Various characterization techniques, including XRD, SEM-EDS, TG, and FTIR, were employed to analyze the steel slag samples before and after carbonation, confirming the formation of calcium carbonate. From a thermodynamic perspective, the sequence of reactivity among the four calcium-based active components in steel slag with $CO_2$ is as follows: $CaO > Ca(OH)_2 > 2CaO·SiO_2 > CaO·SiO_2$.

(3) The direct aqueous carbonation process of steel slag can be divided into two stages: in the initial stage, the rate-limiting step is the mass transfer of $CO_2$; as time progresses, the mass transfer of $Ca^{2+}$ becomes the controlling factor for the carbonation rate.

(4) $CO_2$ sequestration through the direct aqueous carbonation of steel slag as depicted in this study will consume a large amount of fresh water. In the follow-up research process, seawater can be used as a substitute for fresh water, as the reaction medium and its impacts on $CO_2$ mineralization using steel slag needs further research.

**Author Contributions:** Conceptualization, Y.L.; Methodology, Y.L. and L.Z.; Investigation, L.Z.; Data curation, C.L., G.W. and Z.W.; Writing—original draft, F.Z. and G.W.; Writing—review & editing, L.C. and Z.W.; Visualization, J.H.; Supervision, R.X., B.J. and Z.W. All authors have read and agreed to the published version of the manuscript.

**Funding:** This research was funded by China Petrochemical Corporation Scientific Research Projects grant number 122713.

**Institutional Review Board Statement:** Not applicable.

**Data Availability Statement:** The data presented in this study are available on request from the corresponding author. The data are not publicly available due to privacy.

**Conflicts of Interest:** The authors declare no conflict of interest. The funding sponsors had no role in the design of the study; in the collection, analyses, or interpretation of data; in the writing of the manuscript, and in the decision to publish the results.

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
