# Peer review of "Experimental Investigation and Mechanism Analysis of Direct Aqueous Mineral Carbonation Using Steel Slag"

_sustainability, doi:10.3390/su16010081_

Round 1

Reviewer 1 Report

Comments and Suggestions for Authors

The experimental study and mechanism analysis of the carbonization properties of three kinds of steel slag were carried out, which is of great significance to guide the recycling of steel slag waste and reduce carbon emissions, and provides a sustainable development strategy for steel companies and other high-carbon emission enterprises. The quality of the results and arguments is at a good level. The content of the paper is well-structured, the conclusions and references are readable, and the paper as a whole meets the publication requirements, but there are still some minor issues that need further revision, and the comments mentioned below should be considered and the manuscript should be further improved before submitting the final version.

1. line 168 “Equations (2)” should be “Equations (1)”.

2. line 390-398, the format of the equations and the harmony of the text could be improved.

3. The format of Table 3 could be improved.

Author Response

Dear reviewer

Enclosed please find an electronic submission of a revised manuscript and entitled “Experimental Investigation and Mechanism Analysis of Direct Aqueous Mineral Carbonation using Steel Slag” by Fuxia Zhu, Longpeng Cui, Yanfang Liu, Liang Zou, Jili Hou, Chenghao Li, Ge Wu, Run Xu, Bo Jiang*, Zhiqiang Wang* (submitting and corresponding author), which we are submitting for exclusive publication as a research article in Sustainability.

We are truly grateful to you and the reviewer for the critical and valuable comments and thoughtful suggestions. Based on these comments and suggestions, we have made careful modifications on the original manuscript. All changes have been marked in green in the revised manuscript. We hope the new modified manuscript will meet the journal’s standard. Our point-by-point responses to the reviewers’ comments/questions are set forth in the attached file.

Reviewer 2 Report

Comments and Suggestions for Authors

CO2 sequestration is one of the key techniques in the 21 centuries for sustainable development. This manuscript explored the carbonation performance of steel slag at low temperatures. Effects of slag characteristics and operational parameters on the carbonation process, as well as the direct aqueous carbonation mechanism were studied. The manuscript is well-structured and the findings provide a viable technique for CO2 capture and reutilizing waste steel slag. I recommend accepted for publication before some minor revise. My comments are listed below.

(1)   Statistics in Lines 50-53 like 15%, 10-15%, etc. should give references (For example, https://doi.org/10.1016/j.cscm.2021.e00621 ). There are many others; please complement the references in the introduction.

(2)   Lines 147-154, I recommend the authors describe the designing experimental program using a table.

(3)   Section 2.4 Sample Characterization, the experiment details such as sample preparation, testing procedures, set-up parameters, should be given. 

(4)   Figure 7, the presentation form of x-axis value (i.e., 180-150) is inappropriate, what is the value for minor tick marks? 

Author Response

(The authors gave the same response as above.)

Reviewer 3 Report

Comments and Suggestions for Authors

In this paper, researchers evaluated the carbonation performance of three types of steel slag at temperatures below 100℃ using a gas-liquid-solid reaction system. The slag 18 with the highest CO2 sequestration capacity was chosen for a systematic evaluation. A few minor revisions are listed below:

1. "onset" is changed to "beginning" in line 1 of the first paragraph of the introduction.

2. The illustrations in this paper appear somewhat irregular, especially in the eighth page.

3. Table 4 is best placed on the same page.

4. The blank space left on page six is not very pleasing to the eye.

5. The blank space left on page thirteen is also not very pleasing to the eye.

Comments on the Quality of English Language

In this paper, researchers evaluated the carbonation performance of three types of steel slag at temperatures below 100℃ using a gas-liquid-solid reaction system. The slag 18 with the highest CO2 sequestration capacity was chosen for a systematic evaluation. A few minor revisions are listed below:

1. "onset" is changed to "beginning" in line 1 of the first paragraph of the introduction.

2. The illustrations in this paper appear somewhat irregular, especially in the eighth page.

3. Table 4 is best placed on the same page.

4. The blank space left on page six is not very pleasing to the eye.

5. The blank space left on page thirteen is also not very pleasing to the eye.

Author Response

(The authors gave the same response as above.)

Reviewer 4 Report

Comments and Suggestions for Authors

Dear reviewers, please, see the attached file.

Comments on the Quality of English Language

English is fine.

Author Response

Dear reviewer

Enclosed please find an electronic submission of a revised manuscript and entitled “Experimental Investigation and Mechanism Analysis of Direct Aqueous Mineral Carbonation using Steel Slag” by Fuxia Zhu, Longpeng Cui, Yanfang Liu, Liang Zou, Jili Hou, Chenghao Li, Ge Wu, Run Xu, Bo Jiang*, Zhiqiang Wang* (submitting and corresponding author), which we are submitting for exclusive publication as a research article in Sustainability.

We are truly grateful to you and the reviewers for the critical and valuable comments and thoughtful suggestions. Based on these comments and suggestions, we have made careful modifications on the original manuscript. All changes have been marked in green in the revised manuscript. We hope the new modified manuscript will meet the journal’s standard. Our point-by-point responses to the reviewers’ comments/questions are set forth in the attachment file.

Reviewer 5 Report

Comments and Suggestions for Authors

This study initiates an investigation into the carbonation performance of diverse steel slag types under low-temperature conditions. It systematically examines the influence of particle size, temperature, initial CO2 pressure, liquid-to-solid ratio, and stirring speed on the carbonation performance of steel slag. The paper exhibits a commendable structural organization. However, pre-publication attention is warranted for addressing the following concerns.

1.     The data in Table 1 should retain a consistent number of decimal places.

2.     In Line 168, the formula should be numbered as (1).

3.     In Line 228, “whereas the weight loss between 550–950°C experiences a significant increase,” The authors' expression lacks precision; there is an increase between 550 and 850°C, but the increment becomes nearly negligible beyond 850°C. It is recommended that the authors reanalyze the pattern of this curve.

4.     In Line 262, Figure 5, subfigure c, lacks a caption. In Line 387, Figure 11, subfigure a, is missing a caption.

5.     Table 2 is described in the text, and its content is relatively straightforward; therefore, it is recommended to eliminate Table 2.

6.     In Figure 11a, the authors should discuss the slight decrease in stirring speed between 200 and 400. What might be the reasons for this?

7.     In Figure 11b, the vertical axis unit "Mpa" should be "MPa."

8.     The author can briefly outline the limitations of this study and identify issues that require further investigation.

Author Response

(The authors gave the same response as above.)

Round 2

Reviewer 4 Report

Comments and Suggestions for Authors

Dear authors, thank you for your revision. All my comments were properly addressed, and I am now satisfied with the quality of the paper, which is ready for publication. Good job!

I have no other corrections.

Best regards,

The reviewer.

Comments on the Quality of English Language

None.